Insights into the interactions between Deverra tortuosa and Schizomyia buboniae: phytochemicals, antioxidant capacity, and enzyme inhibitory effects

Mahmoud Nashaat N. nashaat_mahmoud@azhar.edu.eg 1
Alqahtani Abdulaziz R. arabe@ub.edu.sa 2
Alotaibi Noura J. 3
Haggag Muhammad I. 1
Nowwar Abdelatti I. 1
Ragab Sanad H. 4
1 Botany and Microbiology Department, Faculty of Science, Al-Azhar University , Cairo , Egypt
2 Department of Biology, College of Science, University of Bisha , Bisha , Saudi Arabia
3 Department of Biology, Faculty of Science, Taif University , Taif , Saudi Arabia
4 Department of Zoology and Entomology, Faculty of Science, Al-Azhar University , Cairo , Egypt
Banaszak Anastazia
Electronic publication date: 2025 Nov 27
Publication date: 2025
Volume: 13
Electronic Location ID: e20052
Received 2025 May 5; Accepted 2025 Aug 17
Copyright: ©2025 Mahmoud et al.
Copyright year: 2025
Copyright holder: Mahmoud et al.
License: This is an open access article distributed under the terms of the Creative Commons Attribution License, which permits unrestricted use, distribution, reproduction and adaptation in any medium and for any purpose provided that it is properly attributed. For attribution, the original author(s), title, publication source (PeerJ) and either DOI or URL of the article must be cited.
License URL: https://creativecommons.org/licenses/by/4.0/

Keywords: Deverra tortuosa (Desf.) DC., Schizomyia buboniae, Insect plant interaction, Phytochemical, Antioxidant

Funding: Graduate Studies and Scientific Research at University of Bisha, KSA The Deanship of Graduate Studies and Scientific Research, University of Bisha This research received external funding from Graduate Studies and Scientific Research at University of Bisha, KSA. The Fast-Track Research Support Program at the Deanship of Graduate Studies and Scientific Research, University of Bisha, supplied funding for this work. The funders had no role in study design, data collection and analysis, decision to publish, or preparation of the manuscript.

==============================
Schizomyia buboniae (Diptera: Cecidomyiidae) induces berry-like galls on the stems of Deverra tortuosa (Desf.) DC. It is also known as “Shabat El-Gabal” and is one of the most important aromatic medicinal plants in Egypt. Many researchers have reported the relationship between galling insects and plant secondary metabolites, but this relationship is not quantitative. This study investigated the impact of S. buboniae-induced galls on physiological traits, phytochemical profiles, antioxidant capacity, and antinutrient levels in D. tortuosa stems. Our results reported that photosynthetic pigment levels, including chlorophyll a, chlorophyll b, total chlorophyll, and carotenoids significantly decreased in galled stems by 63%, 14%, 44%, and 53%, respectively (p < 0.05). Antioxidant enzyme activities such as polyphenol oxidase (PPO), peroxidase (POX), catalase (CAT) and superoxide dismutase (SOD) significantly increased in galled stems by 173%, 88%, 125% and 25%, respectively, indicating elevated oxidative stress response. The analysis of phytochemical compositions revealed that the galled stems of D. tortuosa contained significantly higher levels of total flavonoids, flavonols, saponins, steroids, tannins, alkaloids, cardiac glycosides, and total phenolic compounds compared to non-galled stems, suggesting enhanced production of secondary metabolites. Additionally, galled stems exhibited higher levels of phytates, oxalates, and cyanogenic glycosides than non-galled stems. Proximate contents, including lipids, carbohydrates and proteins, were also elevated in galled stems. Furthermore, galled stems exhibited significantly (p < 0.05) stronger antioxidant activity than non-galled stems. S. buboniae appears to modify the phenotype of D. tortuosa, inducing tissue differentiation and activating defense-related responses. These results reveal that gall composition alters key physiological and biochemical traits in D. tortuosa, possibly as a defense response or as a result of insect interference. The study provides novel insights into the complex plant-insect interaction and highlights the potential implications for the plant medicinal value and suggests that gall-induced tissues of D. tortuosa may be valuable sources of bioactive compounds for pharmaceutical applications.

Introduction

Plant galls are abnormal outgrowths that form when certain organisms stimulate plant tissues. This stimulation causes accelerated cell division and tissue differentiation. More than 2,000 organisms can form galls; animals that can form galls include nematodes, mites, and insects, while microorganisms that can form galls include bacteria, fungi, and viruses. Almost all species of higher plants are capable of producing galls. The phenomenon known as insect galls, which develop when plant tissue is stimulated by an insect’s oviposition or feeding (Hirano et al., 2020). Gall-causing insects can manipulate plant growth and chemical characteristics to promote its development. They alter the phenotype of the host plant by modifying phytochemical pathways, stimulating abnormal tissue growth, and influencing its defense responses. When phytochemicals are manipulated in galling sites, the plant’s defense response against herbivore infestation is restored and tissue growth and differentiation are induced. Thus, the interaction between plants and herbivores induces tissue growth and defense against herbivore infestation. This results in a chemical arms race between plants and herbivores, with phytochemicals playing a dual role in promoting tissue growth and executing endogenous defense (Roy et al., 2022). There is considerable debate about the chronology and causes of the origin or establishment of plant-insect interactions. However, everyone agrees that the evolution of these diverse interactions has led to the diversification of plant and insect species (Kasting & Catling, 2003). Several factors, including climate, geography, and species abundance/distribution, may have contributed to the mutual evolution of plant-insect interactions in time (Mitter, Farrell & Futuyma, 1991; Scott, Stephenson & Chaloner, 1992). Climate change has affected, transformed and fragmented the taxonomic composition as well as the geographic distributions of plants and insects, and is the main driver of the evolution of plant-insect interactions (McElwain & Punyasena, 2007; Wilf, 2008).

The result of the parasitic relationship between insects and plants is insect gall. Beneficiaries are insects: they can evade predators and unfavorable weather conditions because they receive food and a home from the host plant. This interaction slows down plant growth, resulting in decreased plant height and leaf area. Moreover, fatalities can result from severe cases (Fay, Hartnett & Knapp, 1996). However, plant galls typically only cause minor harm to plants, and in some ways, the development of plant galls is a form of self-defense. Galls contain significantly more fat, protein, starch, and tannic acid than other typical plant tissues because they are an aberrant proliferative tissue (Wang et al., 2023).

Approximately 2% of all known insect species are gall-forming species (Dreger-Jauffret, 1992; Labandeira, 2021). Cecidomyiidae (Diptera: Nematocera) is the most diverse, largest, and newest gall midge subfamily, with over 6,651 species (Gagné & Jaschhof, 2004; Gagné & Jaschhof, 2014). The Cecidomyiidae family (gall midges) is one of the most diverse families within the Diptera order, comprising more than 6,600 described species. Due to its widespread distribution and host specificity, this family represents an important component of insect biodiversity. The animal kingdom includes more than 1.8 million described species worldwide, making it one of the most diverse biological kingdoms (Hebert et al., 2016). Insects that produce galls are crucial to the pollination process (Borges, 2015). Certain species of Cecidomyiidae gall midges are economically significant and have the potential to harm crops (Darvas, Skuhrava & Andersen, 2000) as well as forest trees (Skuhravá & Roques, 2000). From India to Algiers, certain galls are utilized for tanning and medicinal purposes due to their tanning properties (Gerling, Kugler & Lupo, 1976). One genus of gall midges is called Schizomyia. Its distribution is characterized as cosmopolitan (Hebert et al., 2016; Borges, 2015; Darvas, Skuhrava & Andersen, 2000). In this work, Gagné provides a comprehensive overview of gall midges in the Neotropical region, including taxonomic descriptions and host plant associations. Schizomyia buboniae is mentioned as one of the species within the genus Schizomyia, which is known for forming galls on specific host plants (Skuhravá & Roques, 2000). In the habitat of the coastal desert, Schizomyia buboniae (Frauenfeld, 1859) causes Deverra tortuosa stems to develop galls that resemble berries (Skuhravá, Skuhravý & Elsayed, 2014).

In the last few years, there has been a rise in the recognition and application of natural bioactive compounds found in aromatic plants (Souto et al., 2021). It’s common knowledge that aromatic plants are regarded as safe substitutes for microbiological resistance and as effective means of managing and controlling pests (Souto et al., 2021; Copping & Duke, 2007).

Deverra tortuosa (Apiaceae) is an aromatic medicinal plant known for its rich content of bioactive compounds. Its interaction with the gall-inducing insect Schizomyia buboniae provides a valuable model for studying plant–insect dynamics and stress-induced phytochemical changes.

Another name for Deverra tortuosa (Desf.) DC. is “Shabat El-Gabal”. It is listed as a common associate in the majority of plant communities found in inland and coastal desert environments (Serag, Khedr & Amer, 2020). Having striate caduceus leaves and dichotomously branched stems, it is a fragrant, glabrous shrub (Boulos, 1995). The plant has many uses, including fuelwood, food, medicine, and aromatic qualities. It also tastes very good to livestock, especially camels (Bedair et al., 2020). It is regarded as an essential range plant in the summer. Its shoots are used as a seasoning and to ease stomach cramps and asthma (El-Seedi et al., 2013). One more critical wellspring of anti-microbials against a few pathogenic microorganisms is the plant’s essential oil (Azzazi et al., 2015). D. tortuosa is among Egypt’s most important medicinal plants. In traditional medicine, it is used as a diuretic, analgesic, carminative, and antiasthmatic for bites, fever, headaches, constipation, and hypertension (Azzazi et al., 2015; El-Mokasabi, 2014). D. tortuosa extract that used in traditional medicine in the treatment of stomach pain and intestinal parasites (Azzazi et al., 2015). The Deverra extracts demonstrated strong scavenging and antioxidant capabilities (El-Lamey, 2015). D. tortuosa aqueous extracts include proteins, terpenoids, alkaloids, tannins, phenolic compounds, and coumarins (Abd El-Moaty et al., 2021). Numerous investigators, including Ahmed, Wassel & Abdel-Moneim (1969), Mahran et al. (1989) and Elmosallamy et al. (2021) have discovered numerous significant phytochemicals with biological activity in various D. tortuosa parts, including flavonoids, unsaturated sterols, coumarins, glycosides, and essential oils. Alkaloids, flavonoids, phenols, tannins, and terpenes are examples of secondary metabolites that play a significant role in protecting plants from herbivores and controlling plant-to-plant interactions through allelopathic action (Kovačević, 2004).

The best material to study the coevolution of plants and insects is plant galls, which are aberrant outgrowths of plant tissues brought on by gall-inducing insects. Gall-inducing insect’s effects on their host plants can be used to shed light on the interactions between the two species as well as the host plant’s growth cycle. Although previous studies have explored the roles of galled inducers and secondary metabolites in host plants, there is a lack of comprehensive data comparing the physiological, hormonal, enzymatic, phytochemical, anti-nutritional, and antioxidant changes between galled and non-galled tissues in D. tortuosa caused by S. buboniae. To our knowledge, no study to date has performed a comprehensive analysis of these variables in this specific plant-insect interaction. Therefore, the present study aims to evaluate the effect of S. buboniae—induced galls on D. tortuosa by analyzing changes in photosynthetic pigments, endogenous hormones, antioxidant enzymes, antinutrients and both primary and secondary metabolites, thus contributing to a better understanding of the physiological and biochemical changes induced by galls in this medicinal plant.

Materials and Methods

Plant gall collection

In February 2024, galled and non-galled stems of Deverra tortuosa (Desf.) DC. were collected from El-Sadat City, Menoufia governorate, Egypt (Fig. 1) (30°22′ 51.96″N, 30°31′35.76″E). Sampling was conducted at three ecological sites. At each site, five galled and five non-galled stems of D. tortuosa were randomly selected. The plant was in its flowering stage and grew in arid conditions characterized by sandy, saline soil. The samples were collected, put in plastic bags and transferred to faculty of Science, Al-Azhar University, Nasr City, Cairo, Egypt. The galled and non-galled stems (Fig. 2), were divided, cleaned, and left to air dry at room temperature in the shade. An electric blender was then used to grind them into a fine powder. The powdered samples were stored for antinutrients, phytochemical and antioxidant analyses in tightly sealed bags.

Figure 1 Map showing samples collections of Deverra tortuosa (Desf.) DC. in Egypt.

Figure 2 (A) Deverra tortuosa (Desf.) DC., (B) Gall shape of Schizomyia buboniae associated with Deverra tortuosa plant.

Quantitative determination of photosynthetic pigments, phytohormone, enzymes activities, proximate, phytochemicals, and anti-nutrient measurements in galled and non-galled stems

Photosynthetic pigments

Photosynthetic pigments were extracted from one g of fresh galled and non-galled stems using 100 mL of 80% acetone. The extract was filtered using Whatman No. 1 filter paper. Absorbance was measured at wavelengths of 470, 649, and 665 nm. The content of chlorophyll a, chlorophyll b, total chlorophyll (a + b), and carotenoids was calculated according to the formulas described by Smith (1974) and Vernon & Seely (2014).

Phytohormone concentration

Phytohormones (GA3, ABA, and IAA) were extracted following Hashem (2006), based on the method of Shindy & Smith (1975). Hormone extracts and standards were methylated as described by Vogel (1956) and analyzed using gas chromatography (HP 5890) equipped with a methyl silicone-coated capillary column (HP-130, 30 m × 0.32 mm × 0.25 µm). The oven temperature was raised to 260 °C at 10 °C/min and held for 10 min. Injector and detector temperatures were set at 260 °C and 300 °C, respectively. Carrier gas flow rates were N2: 30, H2: 30, and air: 300 cm/s.

Enzyme activities

Enzyme extraction followed Mukherjee & Choudhuri (1983). The technique of Marklund & Marklund (1974) was used to estimate the activity of superoxide dismutase (SOD). A 3.6 mL of distilled water, 0.1 mL of enzyme, 5.5 mL of a 50 mM phosphate buffer (pH 7.8), and 0.8 mL of three mM pyrogallol dissolved in 10 mM HCl were combined to create a 10 mL solution. A UV spectrophotometer was then used to measure absorbance at 325 nm to track the rate of pyrogallol reduction.

The assay procedures for measuring catalase (CAT) were based on Aebi (1974). A 9.96 mL of a phosphate buffer (pH 7.0) containing H2O2 was combined with 40 µL of crude enzyme extract to create a reaction mixture with a final volume of 10 mL. A 100 mL of 50 mM phosphate buffer was mixed with 0.16 mL of 30% H2O2 to create the hydrogen peroxide solution. Using a UV spectrophotometer set at 250 nm, the change in absorbance of H2O2 over a 60-second period was used to measure catalase activity.

Polyphenol oxidase (PPO) activities were evaluated using the techniques of Kar & Mishra (1976). A 0.1 M catechol solution in a sodium acetate buffer (pH 5.0) served as the substrate. Absorbance measurements were made at 395 nm after the reaction was run for 60 min at 30 °C.

Peroxidase (POX) activities were ascertained using the procedure of Bergmeyer (2012). The reaction mixture was made up of 0.2 mL of enzyme extract, 5.8 mL of a 50 mM phosphate buffer (pH 7.0), and two mL of a 20 mM H2O2 solution. The rate at which pyrogallol increased absorbance was spectrophotometrically measured using a UV spectrophotometer following the addition of two mL of 20 mM pyrogallol. The measurements were made at 25 °C for 60 s at 470 nm.

Crude lipid

Lipid content of galled and non-galled stems of D. tortuosa was estimated by Soxhlet method (Arunachalam, Saravanan & Parimelazhagan, 2011) with slight modification. A 10 g of dry sample was extracted with 250 mL of petroleum ether (60–80 °C) for 6 h. The extract was evaporated to dryness at 102 °C to a constant weight, then cooled in a desiccator and weighed. The fat content was calculated based on the weight difference and expressed as a percentage of the dry weight.

% Crude lipid = [(Weight of flask and extracted lipid–Weight of empty flask)/(Weight of sample taken)] ×100.

Total carbohydrates

Carbohydrate content was determined following the anthrone method described by Chandran, Nivedhini & Parimelazhagan (2013). A 100 mg sample was hydrolyzed with five mL of 2.5 N hydrochloric acid in a boiling water bath for 3 h, then cooled, hydrolyzed with Na2CO3, and diluted to 100 mL with distilled water. After filtration and centrifugation (3000 × g, 10 min), one mL of the upper liquid was reacted with four mL of freshly prepared anthrone reagent (200 mg of anthrone in 100 mL of 95% H2SO4 solution). The mixture was heated for 8 min, cooled, and the absorbance coefficient was measured at a wavelength of 630 nm. The carbohydrate concentration was calculated using a glucose standard curve and expressed as glucose equivalents (mg/100 g) per 100 g of dry weight.

Crude protein

Crude protein content was estimated using the method of Krishna, Sajeesh & Parimelazhagan (2014). A 100 mg sample was homogenized in 10 mL of 0.2 M phosphate buffer, filtered, and centrifuged at 3,000 × g for 10 min. The upper liquid was diluted to 10 mL with distilled water. A one mL of the diluted extract was reacted with five mL of alkaline copper reagent and incubated for 10 min at room temperature. Then, 0.5 mL of Folin-Ciocalteau (1 N) reagent was added. After 30 min in the dark, the absorbance was measured at 660 nm. The protein concentration was calculated using a standard curve for bovine serum albumin (BSA) and expressed as BSA equivalents in mg per 100 g of dry sample. Nutritional value confirmation followed Indrayan et al. (2005).

Total phenolic acids

The amount of phenolics were estimated using the Folin–Ciocalteu method (Chandra et al., 2014; Mahmoud, Khader & Mahmoud, 2024). Briefly, 50 µL of the extract (one mg/mL) or standard solution (0–50 µg/mL) was mixed with 0.2 mL of 0.5 M Folin-Ciocalteau reagent and 0.6 mL of distilled water. After 5 min, 1.0 mL of 8% Na2CO3 solution was added, and the volume was adjusted to 3.0 mL with distilled water. The mixture was incubated in the dark for 30 min, and the absorbance coefficient measured at 760 nm. The results were expressed in mg gallic acid equivalents per gram of dry weight (mg GA eq/g).

Total tannins

The total tannins were assessed using the Folin-Denis spectrophotometer technique created by Makkar (2003) and Ghanem et al. (2025). A 0.5 mL of the extract (one mg/mL) and 0.5 mL of distilled water were added to 100 mg of polyvinyl poly pyrrolidone. The tubes should be incubated at 4 °C for 4 h. After centrifuged for 10 min at 4 °C at 3,000 rpm. Only non-tannin phenolics are found in the upper liquid. Take 100 µL of the phenolic non-tannin and 0.5 mL of the Folin-Ciocalteu reagent (1 N) in triplicates. Then incubate for 5 min. Including the blank, add 2.5 mL of 5% Na2CO3. Mix and incubate for 40 min in the dark. At 725 nm, absorbance was determined with a UV spectrophotometer. The results were expressed in mg tannic acid equivalents per gram of dry weight (mg TA eq/g).

Total flavonoids

Total flavonoids were determined using the method of Aryal et al. (2019). A 500 µL extract (one mg/mL) was mixed with 0.15 mL of 5% NaNO2 and two mL of distilled water. After 5 min of incubation at room temperature, 0.15 mL of 10% AlCl3 was added, followed by two mL of 4% NaOH and five mL of distilled water. The mixture was incubated at 40 °C for 15 min. Absorbance was measured at 510 nm, and results were expressed as mg quercetin equivalents per gram of dry sample (mg Q eq/g).

Total flavonols

The flavonols were measured using AlCl3 colorimetric technique outlined by Miliauskas, Venskutonis & Van Beek (2004). Combine 500 µL (one mg/mL) of the extract with six mL of CH3COONa and two mL of AlCl3 (2%). The mixture was incubated for 2 h at room temperature, and absorbance was measured at 440 nm. Results were expressed as mg rutin equivalents per gram of dry weight (mg RT eq/g).

Total cardiac glycosides

According to Solich, Sedliakova & Karlíček (1992) Buljet’s reagent was used to measure the cardiac glycoside content. Using 80% ethyl alcohol, plant powder (0.1 g) was extracted and filtered. A 0.5 mL of the extract was mixed with 10 mL of freshly made Baljet’s reagent. A spectrophotometer was used to measure the mixture’s color at 495 nm after it had been diluted with 20 mL of distilled water for one hour. The cardiac glycoside content was determined and expressed as mg/100 g.

Total alkaloids

The total alkaloids were quantitatively determined using the method outlined by Herborne (1973). A five g of the dried sample was combined with a mixture of glacial acetic acid: 70% ethanol (1:4 v/v). After 4 h of incubation, the mixture was filtered. Add a strong ammonia solution to precipitate the alkaloid found in the supernatant. After filtering through pre-weighed filter paper, the alkaloids were dried at 70 °C in an oven until they reached a constant weight. After calculation, the alkaloid content was reported in mg/100 g.

Total saponins

Saponins were determined using (Otang, Grierson & Ndip, 2012; Aldalin et al., 2024) . A 100 mL of 20% ethanol was used to extract five g of the sample twice at 55 °C. The samples were then combined and concentrated to 40 mL at 90 °C. Diethyl ether (40 mL) was used to partition the concentrate, and 60 mL of n-butanol was used to further extract the aqueous layer before being rinsed with 5% NaCl. At 60 °C, the finished extract evaporated to a constant weight. The amount of saponin was expressed as mg per 100 g of dry weight.

Total steroids

Steroids were extracted using Herborne (1973). Five grams of sample were analyzed by boiling it in 50 mL of hydrochloric acid for 30 min, then filtering and extracting it with ethyl acetate. The organic layer was dried, treated with concentrated amyl alcohol, and then heated at 100 °C for 5 min. The turbid extract was filtered, cooled in a desiccator, and weighed. The steroid content was calculated as mg/100 g dry weight.

Phytates

For five hours, four g of each sample were submerged in 100 mL of 2% HCl, and the samples were then filtered out. A conical flask was filled with 25 mL of the filtrate and then five mL of an indicator solution containing 0.3% ammonium thiocyanate (NH4SCN). The pH was then changed in accordance with 3.5 by adding 53.5 mL of distilled water. FeCl3 solution was added to the mixture and titrated until the combination took on a stable brownish yellow color, which lasted five minutes (AOAC, 1990). The amount of phytate per 100 g of dry weight of the plant samples was calculated and reported in milligrams (mg).

Oxalates

A 75 mL of 3.0 M sulfuric acid was combined with one g of each powdered sample, for approximately an hour, the combination was persistently blended with a magnetic stirrer and afterward stressed. A 25 mL sample of the filtrate was taken, and it was titrated with a 0.05 M KMnO4 solution at 80 °C until a light pink hue emerged consistently for at least 30 s (AOAC, 1990). The oxalate content was determined and reported in mg per 100 g.

Cyanogenic glycosides

Cyanogenic glycosides were estimated by AOAC (1990). A four g of sample were mixed with two mL orthophosphoric acid and 40 mL distilled water for 24 h to release bound HCN. The extract was distilled with paraffin (anti-foam) and chips (anti-bumping). A five mL of distillate was collected into a flask containing 0.1 g NaOH in 40 mL water, then diluted to 50 mL. A 20 mL aliquot was mixed with one mL of 5% KI and titrated with 0.01 M AgNO3 until a faint permanent turbidity appeared. Results were expressed as mg/100 g dry weight.

In vitro Antioxidant activity

(2, 2-Diphenyl-1-picrylhydrazyl) radical scavenging activity

The procedure identified by Salem et al. (2022) was used to evaluate the DPPH free radical scavenging activity of the samples. Freshly prepared (0.004%w/v) methanol solution of 2,2 diphenyl-1-picrylhydrazyl (DPPH) radical was prepared and stored at 10 °C in the dark. A three mL of the DPPH solution was mixed with 100 µL (1.95–1000 µg/mL) each of the plant fraction/standard drug (Ascorbic acid). A control-containing DPPH solution and methanol only was also prepared. The reaction mixture was thoroughly vortexed and left to stand in the dark at room temperature for 30 min. Absorbance of the mixture was measured at 517 nm using a UV-visible spectrophotometer (Milton Roy, Spectronic 1201) blanked with methanol. The experiment was done in triplicate. The scavenging ability of the different plant fractions and standard was calculated using the equation:

DPPH Scavenging activity (%) = [(Absorbance of control–Absorbance of sample)/(Absorbance of control)] ×100.

Ferric reducing antioxidant power

The procedure described by Benzie & Strain (1996) was employed to calculate the sample ferric ion reducing antioxidant power. A mixture containing 2.5 mL of 0.2 M phosphate buffer pH 6.6 and 2.5 mL 1% potassium ferricyanide was added to 0.2 mL of the different solvent extracts and standards (50–300 µg/mL). The resulting mixture was incubated for 20 min at 50 °C. After incubation, 2.5 mL of 10% TCA (w/v) was added to terminate the reaction and centrifuged for 10 min at 3000 rpm. 2.5 mL of the supernatant was removed and mixed with 2.5 mL of distilled water and 0.5 ml of 0.1% freshly prepared of FeCl3 was added. The mixture was left to stand for 10 min, and the absorbance was read at 700 nm. A mixture of the buffer instead of the sample served as control. Increased absorbance of the reaction mixture indicated higher reducing power of the plant fractions. The percentage inhibition of the sample and the standard drug was calculated using the formula:

Scavenging activity (%) = [(Absorbance of sample–Absorbance of control)/(Absorbance of highest standard–Absorbance of control)] ×100.

Hydrogen peroxide scavenging activity

Hydrogen peroxide (H2O2) scavenging activity was assessed using Ruch, Cheng & Klaunig (1989). A 100 µL were mixed with 600 µL of two mM phosphate buffer (pH 7.4) at different concentrations (50–300 µg/mL), and the volume was adjusted to four mL. After 10 min of incubation, the absorbance was measured at 230 nm. Scavenging activity (%) was calculated using:

Scavenging activity (%) = [(Absorbance of control–Absorbance of sample)/(Absorbance of control)] ×100.

Reducing antioxidant power

Reducing antioxidant power was estimated using Oyaizu’s method (Oyaizu, 1986). Various concentrations of each sample and standard (50–300 µg/mL) were prepared in methanol. To each tube, one mL aliquot was mixed with 2.5 mL of 0.2 M phosphate buffer (pH 6.6) and 2.5 mL of 1% potassium ferricyanide were added. After incubation at 50 °C for 20 min, 2.5 ml of 10% trichloroacetic acid was added and centrifuged at 650 ×g for 10 min. Then, 2.5 mL of the upper liquid was mixed with 2.5 mL of distilled water and 0.5 mL of 0.1% ferrous chloride. The absorbance was measured at a wavelength of 700 nm.

Scavenging activity (%) = [(Absorbance of sample–Absorbance of control)/(Absorbance of highest standard–Absorbance of control)] ×100.

Statistical analysis

Minitab® version 19 and Microsoft Excel version 365 were used for statistical studies using analysis of variance (ANOVA) (Tukey’s Honestly Significant Difference (HSD)) and t-test at the 0.05 level of probability. Parametrically distributed quantitative data was gathered. An acceptable margin of error of 5% and a 95% confidence interval were determined.

Results

Photosynthetic pigments

The results for photosynthetic pigments in Fig. 3 clearly show a significant reduction in chlorophyll a, chlorophyll b, total chlorophyll (a + b), and carotenoid contents in the galled stems of Deverra tortuosa compared to the non-galled stems. The chlorophyll (a), (b), (a + b) and carotenoids significantly decreased by approximately 63%, 14%, 44%, and 53%, respectively, compared to the non-galled tissues (P < 0.05).

Figure 3 The chlorophylls (a, b, a + b) and carotenoids contents (mg/g of fresh weight) in galled and non-galled stems of D. tortuosa.

Data in the columns are presented as mean ± deviation, (n = 3). Values with different superscript letters indicate statistically significant differences based on Tukey’s HSD test (p < 0.05). Superscript letters (a, b) denote statistically significant differences between groups in pairwise comparisons.

Phytohormone concentration

The findings of the current study, as shown in Fig. 4, indicate a significant increase in the contents of indole-3-acetic acid, gibberellic acid, and abscisic acid in the galled stems of D. tortuosa compared to the non-galled stems. Specifically, the levels of GA3, ABA, and IAA increased by approximately 133%, 236%, and 590%, respectively.

Figure 4 The Phytohormones contents (mg/100 g of fresh weight) in galled and non-galled stems of D. tortuosa.

Data in the columns are presented as mean ± deviation, (n = 3). Values with different superscript letters indicate statistically significant differences based on Tukey’s HSD test (p < 0.05). Superscript letters (a, b) denote statistically significant differences between groups in pairwise comparisons.

Enzyme activities

The obtained data (Fig. 5) indicated that enzyme activities were significantly increased in the galled stems compared to non-galled stems, with polyphenol oxidase (PPO) increasing by about 173%, peroxidase (POX) by 88%, catalase (CAT) by 125%, and superoxide dismutase (SOD) by 25% (P < 0.05).

Figure 5 The enzyme activities (unit/g F.wt./hour) of POX, CAT, SOD, and PPO in galled and non-galled stems of D. tortuosa.

Data in the columns are presented as mean ± deviation, (n = 3). Values with different superscript letters indicate statistically significant differences based on Tukey’s HSD test (p < 0.05). Superscript letters (a, b) denote statistically significant differences between groups in pairwise comparisons.

Proximate, phytochemicals, and anti-nutrient

The results for the proximate analysis of the galled and non-galled stems of D. tortuosa, are summarized in Table 1. Glucose was used as a standard to estimate total carbohydrates. The total carbohydrate content of galled stems (28.11 ± 1.75 g glucose/100 g) was higher than that of non-galled stems (24.36 ± 2.91 g glucose/100 g). Bovine serum albumin (BSA) was used as a standard to estimate total soluble protein. The total protein content of galled stems (12.08 ± 2.98 g BSA/100 g) was higher than that of non-galled stems (8.43 ± 1.11 g BSA/100 g). The crude lipid content of galled stems (1.67 ± 0.31 mg/100 g) was higher than that of non-galled stems (1.23 ± 0.35 mg/100 g). The nutritional value of galled stems was 175.74 ± 3.8 kcal/100 g, which was significantly higher (P < 0.05) than that of non-galled stems (142.24 ± 9.91 kcal/100 g).

The phytochemical composition of galled and non-galled stems of D. tortuosa is shown in Table 1. The total phenolic content of galled stems (91.62 ± 7.29 mg GA eq/g) was higher than that of non-galled stems (82.85 ± 5.5 mg GA eq/g). The total flavonoids and flavonols of galled stems (258.93 ± 10.01 mg Q eq/g, 200.60 ± 16.12 mg RT eq/g, respectively) were higher than that of non-galled stems (234.93 ± 26.11 mg Q eq/g, 150.10 ± 2.5 mg RT eq/g, respectively). The total tannins of galled stems (50.93 ± 2.08 mg TA eq/g) were higher than that of non-galled (45.93 ± 7.22 mg TA eq/g).

The total cardiac glycoside content in galled stems (64.64 ± 5.8 mg/100 g) was significantly higher than that of non-galled stems (37.98 ± 2.22 mg/100 g). The levels of total alkaloids, saponins, and steroids were also significantly higher (P < 0.05) in galled stems compared to non-galled stems, with respective values of 2.02 ± 0.34, 0.96 ± 0.07, and 3.52 ± 0.71 mg/100 g in galled stems, versus 1.09 ± 0.02, 1.37 ± 0.18, and 0.72 ± 0.12 mg/100 g in non-galled stems.

Table 1 Quantification of proximate, phytochemical, and anti-nutrients composition in galled and non-galled stems of D. tortuosa.

	Components	Galled stems	Non-galled stems	p-value	
Proximate composition	Carbohydrates (g glucose/100 g)	28.11 ± 1.75	24.36 ± 2.91	0.152	
Proteins (g BSA /100 g)	12.08 ± 2.98	8.43 ± 1.11	0.185	
Lipids (mg /100 g)	1.67 ± 0.31	1.23 ± 0.35	0.205	
Nutritive value (Kcal /100 g)	175.74 ± 3.8	142.24 ± 9.91	0.032	
Phytochemical contents	Phenolic acids (mg GA eq/g)	91.62 ± 7.29	82.85 ± 5.5	0.199	
Tannins (mg TA eq/g)	50.93 ± 2.08	45.93 ± 7.22	0.368	
Flavonoids (mg Q eq/g)	258.93 ± 10.01	234.93 ± 26.11	0.275	
Flavonols (mg RT eq /g)	200.60 ± 16.12	150.10 ± 2.5	0.033	
Cardiac glycosides (mg /100 g)	64.64 ± 5.8	37.98 ± 2.22	0.018	
Alkaloids (mg /100 g)	2.02 ± 0.34	0.96 ± 0.07	0.033	
Saponins (mg /100 g)	3.52 ± 0.71	1.09 ± 0.02	0.027	
Steroids (mg /100 g)	1.37 ± 0.18	0.72 ± 0.12	0.014	
Anti-nutrients	Oxalates (mg /100 g)	31.47 ± 1.62	29.01 ± 3.09	0.31	
Phytates (mg /100 g)	5.86 ± 0.73	3.67 ± 0.72	0.034	
Cyanogenic glycosides (mg /100 g)	17.01 ± 1.5	15.66 ± 1.69	0.377	
Notes.

Results (n = 3) are shown as mean ± standard deviation.

BSA Bovine serum albumin

GA eq/g Gallic acid equivalent per gram

TA eq/g Tannic acid equivalent per gram

Q eq/g Quercetin equivalent per gram

RT eq /g Rutin equivalent per gram

The antinutrients content are displayed also in Table 1. Oxalates, phytates, and cyanogenic glycosides were higher in galled stems compared to non-galled stems, recording values of 31.47 ± 1.62, 29.01 ± 3.09, 5.86 ± 0.73, 3.67 ± 0.72, 17.01 ± 1.5, and 15.66 ± 1.69 mg/100 g, respectively.

Antioxidant activity

In the current study, the process of scavenging DPPH radicals of galled and non-galled stems of D. tortuosa in comparison to known antioxidants (ascorbic acid) and half of the maximum amount that inhibits [IC50] of the radicals at various concentrations are presented in Fig. 6 and Table 2, respectively. The scavenging activity of galled stems increased with increasing concentrations of the standard compound. At 31.3 µg/mL, almost all the standard drug, galled, and non-galled stems had over 50% inhibitory activity (64.76%, 57.31%, and 53.82%, respectively) on the process of DPPH scavenging. While, at 1,000 µg/mL the scavenging activity of galled, non-galled and standard drug were 95.35%, 91.48%, and 98.84%, respectively. The IC50 (the ability of galled and non-galled stems of D. tortuosa absorbing or suppressing half of the radicals) was 21.43 µg/mL in galled stems, 27.91 µg/mL in non-galled with comparison to ascorbic acid (15.78 µg/mL). The decreasing scavenging activity of galled, non-galled and the order of the standard medications was determined by the IC50; ascorbic acid > galled stems > non-galled stems (Table 2).

Figure 6 DPPH activity of ascorbic acid (positive control) galled, and non-galled stems of D. tortuosa at different concentrations.

Data in the columns are presented as mean ± deviation, (n = 3). Values with different superscript letters indicate statistically significant differences based on Tukey’s HSD test (p  <  0.05).

The antioxidant potential of galled and non-galled stems of D. tortuosa, depends on their capacity to convert the ferrous tripyridyltriazine (TPTZ - Fe III) complex into the ferric tripyridyltriazine (TPTZ - Fe II) complex in the FRAP assay. The reducing power of galled and non-galled stems of D. tortuosa is gradually increased with an increase in concentration. The final concentration of 300 µg/mL showed the largest relative percentage of reducing power by galled stems (91%), followed by non-galled (76.61%) (Fig. 7). The IC50 obtained for galled stems, non-galled stems, and ascorbic acid was 149.62, 184.40, and 109.11 µg/mL, respectively. An increase in the activity of reducing power of ascorbic acid, galled and non-galled stems as obtained from the IC50 is in the order; ascorbic acid > galled stems > non-galled stems (Table 2).

Stems with insect gall (89.44%) have shown a nearly substantial H2O2 scavenging effect at final concentrations when in contrast to the ascorbic acid reference standard. A moderate degree of effective H2O2 scavenging was shown by non-galled stems (80.31%), demonstrating good scavenging capacity (Fig. 8). Potassium ferricyanide must be converted to the ferrous state by antioxidants to pass the reducing power test, also known as the K3 [Fe (CN)6] complex. The test solution exhibits color variations, ranging from green to blue or Perl’s Prussian blue Fe4 [Fe (CN)6]3. To determine the color of the solution, the concentration of each sample can be determined spectrophotometrically at 700 nm. The reference standard, ascorbic acid, was used to express the reducing power. The galled stems (89.12%) have shown excellent reducing capacity when it comes to potassium ferricyanide (Fig. 9).

Discussion

The aims of this study were to investigate the chemical differences between galled and non-galled tissues of Deverra tortuosa, and to examine whether the gall-forming insect Schizomyia buboniae influences the chemical composition of gall tissue. This was assessed through experimental data on various chemical groups, highlighting the interaction between gall-inducing insects and plant secondary metabolites across different species and environmental conditions. This work focuses on the interaction between gall inducer by S. buboniae and host plant photosynthetic pigments, hormones, enzymes, antinutrients, antioxidants, primary and secondary in D. tortuosa stems with and without galls as a comparative study. Our results suggest that galled stems of D. tortuosa are associated with higher concentrations of specific kinds of physiology, antinutrients, secondary metabolites and antioxidants than those in without galls. This could be because these chemicals have different functions in plant defense. The accumulation of these chemical compounds in the external gall tissue may shield the gall insect from infections, other herbivores, and predators (Cornell, 1983; Hartley & Lawton, 1992). Though our conclusions should be interpreted cautiously due to the small number of studies and issues with meta-analyses, these results shed light on the connection between galling insects and plant chemistry.

Table 2 IC50 values of ascorbic acid, galled and non-galled stems of D. tortuosa.

	DPPH	FRAP	H 2 O 2	RAP	
	IC 50 (µg/mL)	R 2	IC 50 (µg/mL)	R 2	IC 50 (µg/mL)	R 2	IC 50 (µg/mL)	R 2	
Galled stems	21.43	0.9744	149.62	0.9797	118.48	0.9709	138.12	0.9552	
Non-galled stems	27.91	0.9861	184.40	0.9565	138.69	0.9652	168.34	0.9969	
Ascorbic acid	15.78	0.9801	109.11	0.9943	93.70	0.9964	88.71	0.9966	
Notes.

IC50 Scavenge/inhibit 50% of the radical at the concentration (μ g/mL) required

R2 Coefficient of determination: values from regression lines with a 95% degree of confidence

DPPH 2,2diphenyl-1-picrylhydrazyl

FRAP Ferric reducing antioxidant power

H2O2 Hydrogen peroxide

RAP antioxidant power

Figure 7 FRAP activity of ascorbic acid, galled, and non-galled stems of D. tortuosa at different concentrations.

Data in the columns are presented as mean ± deviation, (n = 3). Values with different superscript letters indicate statistically significant differences based on Tukey’s HSD test (p < 0.05).

Figure 8 Hydrogen peroxide scavenging activity of ascorbic acid, galled, and non-galled stems of D. tortuosa at different concentrations.

Data in the columns are presented as mean ± deviation, (n = 3). Values with different superscript letters indicate statistically significant differences based on Tukey’s HSD test (p < 0.05).

Figure 9 Reducing antioxidant power activity of ascorbic acid, galled, and non-galled stems of D. tortuosa at different concentrations.

Data in the columns are presented as mean ± deviation, (n = 3). Values with different superscript letters indicate statistically significant differences based on Tukey’s HSD test (p < 0.05).

Gall-causing insects can alter host plant physiology by disrupting pigment biosynthesis and reprogramming nutrient distribution, consistent with the sink theory, which collectively contribute to decreased photosynthetic efficiency (Mishra, Saini & Patni, 2024). In our investigation, the contents of chlorophyll (a), (b), (a+b) and carotenoids were significantly declined in galled stems when compared to non-galled (Fig. 3). Less photosynthetic pigment is typically seen in plant leaves that have galls (Aytar & Keleş, 2025). This could be due to either an interruption in the pigment’s biosynthesis or an accelerated breakdown of the pigment (Kot, Rubinowska & Michałek, 2018). In addition, leaves with galls have a lower concentration of photosynthetic pigments because the galls themselves serve as sinks for photo-assimilates, draining the leaves of their photo-assimilates during the gall development process (Martini et al., 2020).

The physical effects of gall formation, hormonal imbalances brought on by insect infestation may also be responsible for the observed decrease in photosynthetic pigments, such as carotenoids, total chlorophyll, chlorophyll a, and chlorophyll b, in the galled stems of D. tortuosa. In particular, a physiological basis for pigment degradation is suggested by the simultaneous rise in abscisic acid (ABA) and decrease in gibberellic acid (GA3) found in the same tissues. While ABA encourages the breakdown of chlorophyll under stress, GA3 is known to increase chlorophyll biosynthesis and postpone senescence. Hormonal disruption may contribute to impaired photosynthetic capacity during gall development, as evidenced by the significant decrease in GA3 and increase in ABA in galled stems that correlates with the decline in pigment levels (Murchie & Niyogi, 2011).

In insect-fed plants, the accumulation of certain secondary metabolites plays dual roles: some compounds act as signaling molecules that activate broader plant defense pathways, while others exert direct toxic effects that deter or kill insects that feed on them (Ahammed et al., 2024). They also play a role in the production of highly reactive quinones that are deadly to insects (Yousuf et al., 2024). In our investigation, we discovered that, in comparison to the control, the galled stems had increased SOD, CAT, POX, and PPO enzymes activities. Several plant species have also shown increased antioxidant enzymes activities as a result of feeding on insects (Taggar et al., 2012; Golan, Rubinowska & Górska-Drabik, 2013; Kot et al., 2018).

Insect-induced gall induction usually results in metabolic disruptions and cellular damage in the tissues of the host plant, which increases the production of reactive oxygen species (ROS) like hydroxyl radicals (•OH), hydrogen peroxide (H2O2), and superoxide radicals (O2−). ROS are important signaling molecules that activate stress response pathways in plants, despite the fact that they can also be detrimental (Mittler, 2002; Suzuki et al., 2012). Genes encoding antioxidant enzymes like superoxide dismutase (SOD), catalase (CAT), and peroxidases (POD) are upregulated as a result of this oxidative burst, which also sets off downstream signaling cascades involving mitogen-activated protein (MAP) kinases, calcium-dependent protein kinases (CDPKs), and redox-sensitive transcription factors (Apel & Hirt, 2004; Foyer & Noctor, 2005). The higher activity of these enzymes seen in D. tortuosa stems most likely indicates a protective reaction to oxidative stress brought on by insects. In addition to aiding in the maintenance of redox homeostasis, this enzymatic upregulation might be a component of a feedback mechanism that affects insect performance or gall development, possibly decreasing insect fitness or modifying host metabolism to benefit the gall-former.

Plant cells contain low or high concentrations of the endogenous acidic phytohormones (IAA, GA3, and ABA), which control plant growth and function as signaling regulators to lessen biotic and abiotic stress (Abd-Elsalam & Mohamed, 2024). In our study, elevated levels of IAA and GA3 in galled stems indicate increased cell division and elongation, promoting gall tissue formation. Elevated abscisic acid levels may reflect stress-related signals or modulation of stomatal activity. These hormonal shifts likely indicate a reprogramming of host signaling pathways, possibly regulated by S. buboniae larvae to facilitate gall initiation and development. These results were in harmony with those reported by Tokuda et al. (2013) which found that GA3 and ABA levels in maize plant infected with the gall-inducing Cicadulina bipunctata (Melichar) were ten to thirty times greater than those in control plants that were not affected. Tooker & De Moraes (2011) reported that IAA was found to be highly concentrated in two distantly related taxa that induce gall formation: a lepidopteran (Gnorimoschema gallaesolidaginis) and a dipteran (Eurosta solidaginis). Additionally, these taxa produced even greater IAA concentrations in developing galls, suggesting that larvae oversaw the IAA concentration, which appeared to be crucial for gall formation. These results also were illustrated by Zhu, Chen & Liu (2011) in Triticum aestivum and Oryza sativa L. plants.

Carbohydrates make up the main component of D. tortuosa. Simple sugars, disaccharides, polysaccharides (starch), and a few organic acids were among them. An essential component of both plant and animal cells is carbohydrates. Their physiological roles include protein preservation, energy storage and supply, and inhibition of ketone body synthesis (Jiang et al., 2021). Carbohydrates play a pivotal role in energy metabolism and resource allocation within plants. In the context of gall formation, gall-forming insects alter the plant source sink relationship, transforming gall tissues into powerful metabolic sinks that actively accumulate sugars and starches. This transformation supports larval development by ensuring a continuous nutrient supply (Larson & Whitham, 1991). Studies have shown elevated carbohydrate levels in galled tissues, suggesting that insect induced galls can hijack host transport systems to prioritize sugar delivery to the gall site (Hartley, 1998).

Such changes not only result in insect survival but also reflect a profound reprogramming of the host’s physiological and metabolic state. The level of total carbohydrates of D. tortuosa are similar to the results of Ragab et al. (2024), who found that total carbohydrates in Avicenna marina leaves with galls stimulated by Actilasioptera gagné (Diptera: Cecidomyiidae) (24.61 ± 0.54 g glucose/100 g) were higher than those in leaves non-galls (22.69 ± 0.46 g glucose/100 g).

Plant protein is easily absorbed and digested by the human body, and it has a wide range of nutritional benefits. It performs a number of tasks, including immune regulation, antioxidation, and antifatigue properties (Fang et al., 2022). Protein levels in galled plant tissues often reflect increased metabolic activity and local stress responses resulting from insect infestation. Galls are complex structures characterized by accelerated cell division, differentiation, and nutrient flow, all of which require increased enzymatic and structural proteins (Shorthouse, Wool & Raman, 2005). High protein content may indicate activation of defense related pathways, including disease associated proteins, heat shock proteins, and enzymes involved in mitigating oxidative stress (de Oliveira & dos Santos Isaias, 2010). These differences suggest that gall-causing insects can manipulate host metabolism to create a microenvironment optimized for insect feeding and protection, while also eliciting a localized stress response in the plant. Our findings also have similarities to those of Ragab et al. (2024) who found a protein content of Avicennia marina in the range of 5.99 to 8.5 g BSA/100 g for leaves without galls and with galls, respectively.

Lipids are essential components of plant growth and defense, playing critical roles in membrane structure, energy storage, and signaling pathways. In galled tissues, high lipid content may be associated with increased cell proliferation, as new membranes are essential for rapid cell division (Allison & Schultz, 2005). In addition, fat contributes to the formation of protective barriers, such as waxy skin, which may strengthen gall tissue against external stresses or dehydration. The crude lipid content of D. tortuosa stems with galls was higher than those in non-galled (Table 1). Ragab et al. (2024), also showed 0.072 mg/100 g higher crude lipid content in A. marina more gall-containing leaves than non-galled leaves (0.021 mg/100 g).

Significant ecological and physiological mechanisms linked to gall formation are probably reflected in the variations in lipid and carbohydrate levels between galled and non-galled tissues. Redirecting host resources to support the gall-inducing insect, which controls host metabolism to provide a nutrient-rich microenvironment, may be indicated by a decrease in these main metabolites in galled tissues. The plant’s entire energy balance may be jeopardized, and its ability to grow and reproduce may be diminished (Hartley, 1998). Moreover, decreased levels of fat and carbohydrates could indicate a change in the plant’s resource allocation strategy, moving from primary metabolism to the synthesis of protective secondary metabolites. These modifications could be the insect’s defense mechanism or the outcome of its attempts to inhibit host resistance mechanisms.

Plant polyphenols are a class of compounds that exhibit strong antioxidant properties. They actively prevent dementia and heart disease and have antiviral and antitumor properties (Fraga et al., 2019). In the industries of food, medicine, and cosmetics, they are extensively utilized (de Lima Cherubim et al., 2020). Our results showed excellent agreement with those of Ragab et al. (2024) who discovered the total content of phenols in Avicenna marina leaves with galls (70.42 ± 0.61 mg GAE/g dry weight) were greater than those in leaves without galls (26.90 ± 1.25 mg GAE/g dry weight). Wang et al. (2023) additionally demonstrated that in the seven solvent extracts of Picea koraiensis Nakai with gall inducing by Adelges laricis, the total phenolic content ranged from 0.92 ± 0.03 to 77.11 ± 0.52 mg GAE/g.

Plants of many different kinds contain flavonoids, a class of polyphenolic compounds whose fundamental structural unit is 2-phenylchromone. In addition to controlling blood pressure and cholesterol, inhibiting the growth of cancer cells, delaying aging and cell deterioration, and having potent protective effects against diseases of the heart and brain, flavanoids also play a crucial role as antioxidants and immunomodulatory agents, thereby contributing significantly to human health (Wen et al., 2021). The total flavonoids and flavonols of galled stems (258.93 ± 10.01 mg Q eq/g, 200.60 ± 16.12 mg RTE/g dry weight, respectively) were higher than those in non-galled (234.93 ± 26.11 mg Q eq/g, 150.10 ± 2.5 mg RTE/g, respectively). Similar findings were also previously reported by Ragab et al. (2024). Our results are higher than the results obtained by Slima, Alhobishi & Turki (2021) who found that the total flavonoids in aerial parts of D. tortuosa collected from various locations in Egypt ranged from 10.81 to 73.56 mg/g.

Phloroglucinol, or tannins found in plants, are polyphenols that are abundant in both marine and terrestrial plants. For a long time, plant tannins have been added to animal production (Xiao et al., 2018). Tannin can be used as an antidote for heavy metal and alkaloid poisoning in addition to its antibacterial and antiviral properties. Because of its potent reduced qualities, it can prevent aging by eliminating superoxide free radicals from the body (Tong et al., 2022). This is comparable to the outcomes of Ragab et al. (2024) who found that total tannin content in Avicenna marina leaves with galls (33.85 ± 0.41 mg TAE/g dry weight) were higher than those in leaves without galls (6.68 ± 0.55 mg TAE/g dry weight). Our results are higher than the results obtained by Slima, Alhobishi & Turki (2021) who discovered that the total tannin content of aerial parts of D. tortuosa varied between 5.29 and 17.76 mg/g, respectively.

The elevated levels of flavonoids and tannins in the galled stems of D. tortuosa may indicate an induced defense response triggered by oxidative stress or tissue damage caused by S. buboniae larvae. Flavonoids and tannins are well-known secondary metabolites involved in plant defense against herbivores and pathogens; they can act as feeding deterrents, enzyme inhibitors, or signaling molecules in defense pathways (Treutter, 2006; Barbehenn & Constabel, 2011). However, a number of studies have also demonstrated that insects that cause gall can alter the metabolic pathways of plants to promote the accumulation of particular phenolics, which may inhibit localized immune responses or help the insect mobilize nutrients (Raman, 2011). For instance, some flavonoids may have antioxidant properties that help the insect indirectly by stabilizing gall structure and lowering the host’s defense reactivity. Thus, the accumulation of flavonoids and tannins may be the result of a complex interaction, a combination of host defense and insect induced biochemical reprogramming.

The accumulation of anti-nutritional substances like cyanogenic glycosides, oxalates, and phytates has physiological and ecological significance, especially when it comes to interactions between plants and insects and herbivore deterrence. When concentrations of these substances surpass specific thresholds, toxic effects may result. For instance, livestock may become toxic if oxalate levels exceed 2% of dry matter because insoluble calcium oxalate crystals may form (Savage et al., 2000). Although they are essential for the storage of phosphorus in seeds, phytates can chelate vital minerals such as calcium, iron, and zinc, decreasing their bioavailability, particularly when they are present in excess of 1% (Hurrell, 2003). Cyanogenic glycosides, which release hydrogen cyanide upon tissue damage, are toxic to both insects and vertebrates, with toxic effects typically observed at concentrations above 200 ppm (Poulton, 1990). These compounds are often part of the plant’s chemical defense system and may be regulated in response to weed growth or tannin formation. Based on the toxicity levels mentioned above, the content of these compounds in our results is much lower than the toxicity limit when converted into percentages.

The different antioxidants neoagarooligosaccharides (NOAs) possess various modes of action. These could serve as hydrogen abstractors, chain initiation reaction terminators, peroxide decomposers, free radical scavengers, and binders in catalysts for transition metal ions. Because of this, a single antioxidant assay is unable to determine the entire any plant extract’s potential profile for antioxidants (Niki, 2011; Niki, 2016). Consequently, a variety of antioxidant assays were employed to evaluate each plant extract’s overall antioxidant potential profile. Studies on the antioxidant potential of the medicinal herb D. tortuosa in a variety of human conditions show that free radicals are the cause of these conditions, including Parkinson’s disease, respiratory conditions, diabetes and sclerosis, and cardiovascular issues (Malik et al., 2021). Thus, antioxidant materials are used to decrease the effects of free radicals (Akbari et al., 2022). The main component that possessed the capacity to absorb free radicals was polyphenolics. The presence of a greater content of flavonoids and phenols in stems with insect gall compared to stems without gall accounts for their higher DPPH scavenging potential. Similar findings were also previously reported by Mekky et al. (2024) proved that the IC50 of the aerial flowering parts of Centaurea calcitrapa L. were 2.82, 4.79, 6.33, and 8.03 µg/mL for methanol, ethyl acetate, chloroform, and aqueous extracts, respectively.

Every antioxidant assay produced noteworthy results when considering the goals of the ongoing studies. A common, inexpensive, and simple method for reducing a reaction between chemicals and the DPPH test is used to assess a drug’s potential antioxidant capacity. Because of its wide absorption band, deep violet in solution is the DPPH radical; when neutralized, it turns light yellow or colorless. The response can be visually observed thanks to this feature. Redox-connected colorimetric examine with a total electron move response component utilizes cell reinforcements as reductants in the Ferric reducing antioxidant power (FRAP) test. The extract exhibiting the maximum percentage reduction power in relation was taken from stems that had insect gall. High-oxygen dismutase (Rosini & Pollegioni, 2022) and additional monomeric oxidases, such as xanthine oxidase and amino acid oxidase, that are present in the outer mitochondrial membrane (Birben et al., 2012), may activate the superoxide anion dismutase to generate hydrogen peroxide. Since H2O2 can either create or convert cytotoxic hydroxyl radicals (•OH), even though it is not on its own reactive, it can irregularly be extremely damaging to cells. Heat-globin and other heat-sensitive proteins may be broken down, and metal ions may be oxidized to create superoxide-free radicals like Fe2+ to Fe3+, Cu+ to Cu2+, or what is known as Fenton (Matuz-Mares et al., 2021) and Haber-Weiss mechanisms within cells, developing into incredibly strong oxidizing agents. Here, stems exhibiting insect gall have shown almost significant ability to scavenge H2O2 in comparison to ascorbic acid, as the standard. A sensitive “semi quantitative” measurement of diluted polyphenolics in a redox process is made possible by the Fe3+/ferricyanide system (Akbaribazm, Khazaei & Khazaei, 2020). Additionally, it can be a helpful indicator of an extract’s potential antioxidant rate (Purushothaman et al., 2022). A strong reducing ability was observed in the galled stems of D. tortuosa using the potassium ferricyanide assay.

Conclusion

This research focused on the gall-causing insect S. buboniae and its interaction with D. tortuosa. As a result, the physiological, phytochemical, antioxidant, and nutritional characteristics of infected and uninfected D. tortuosa stems have not been fully investigated. This study highlights significant changes in photosynthetic pigments, hormone levels, antioxidant enzyme activity, and secondary metabolites, offering fresh insight into the physiological and biochemical reactions of D. tortuosa to gall induction by S. buboniae. These results point to a complex interaction between insect-induced metabolic manipulation and plant defense mechanisms. We highlight the ecological significance of these metabolite changes and their possible roles in plant–insect coevolution rather than making conjectural claims about pharmaceutical applications. To determine their pharmacological significance, future studies should examine the specific bioactivities of the modified compounds using targeted assays, such as cytotoxic, enzyme inhibition, or antimicrobial tests; transcriptomic or proteomic analyses could also help clarify the regulatory networks underlying gall-induced metabolic reprogramming in D. tortuosa.

Supplemental Information

Supplemental Information 1 Raw data proximate, phytochemical and antioxidants

Supplemental Information 2 Raw data chlorophyll and hormones

Supplemental Information 3 Raw data enzymes

Additional Information and Declarations

Competing Interests

Author Contributions

Data Availability

The authors declare there are no competing interests.

Nashaat N. Mahmoud conceived and designed the experiments, performed the experiments, analyzed the data, prepared figures and/or tables, authored or reviewed drafts of the article, and approved the final draft.

Abdulaziz R. Alqahtani analyzed the data, authored or reviewed drafts of the article, and approved the final draft.

Noura J. Alotaibi analyzed the data, prepared figures and/or tables, and approved the final draft.

Muhammad I. Haggag performed the experiments, authored or reviewed drafts of the article, and approved the final draft.

Abdelatti I. Nowwar conceived and designed the experiments, performed the experiments, prepared figures and/or tables, and approved the final draft.

Sanad H. Ragab conceived and designed the experiments, analyzed the data, authored or reviewed drafts of the article, and approved the final draft.

The following information was supplied regarding data availability:

The raw data is available in the Supplementary Files.

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
