# Peer review of "Insights into the interactions between Deverra tortuosa and Schizomyia buboniae: phytochemicals, antioxidant capacity, and enzyme inhibitory effects"

_PeerJ, doi:10.7717/peerj.20052_

## Round 0.1 · original submission · Major Revisions

· Academic Editor

Major Revisions

We received evaluations of your manuscript from three expert reviewers and their comments can be seen below. All agree that this is an important contribution, however all have pointed out some major and minor points that need to be taken into account in a revised version of the manuscript. Please ensure that you clearly inidcate how and where you have incorporated these concerns into the manuscript. This will ensure a smooth revision process

**Language Note:** PeerJ staff have identified that the English language needs to be improved. When you prepare your next revision, please either (i) have a colleague who is proficient in English and familiar with the subject matter review your manuscript, or (ii) contact a professional editing service to review your manuscript. PeerJ can provide language editing services - you can contact us at [email protected] for pricing (be sure to provide your manuscript number and title). – PeerJ Staff

Reviewer 1 ·

Basic reporting

-

Experimental design

-

Validity of the findings

-

Additional comments

In this study, the authors examined the impact of Schizomyia buboniae-induced galls on the biochemical, physiological, and antioxidant properties of Deverra tortuosa stems by comparing infested and non-infested tissues. They reported enhanced levels of secondary metabolites, enzymatic activities, phytohormones, and antioxidant capacity in the galled tissues, suggesting potential pharmaceutical relevance. Although the study presents interesting findings on the interaction between S. buboniae and D. tortuosa, and the data suggest potential bioactivity in gall-induced plant tissues, the manuscript requires major revisions before it can be considered for publication. Below, I provide detailed comments and suggestions from the abstract to the conclusion, organized by manuscript line numbers.

Abstract
Line 26: "This relationship is not quantitative." – Please clarify what is meant by "not quantitative." Do the authors mean that prior studies lacked measurable comparisons or statistical rigor?
Line 31: "increased concentrations of physiology" – This phrase is unclear. Physiology is not a concentration. Please revise to specify which physiological parameters were increased.
Line 35: Sentence structure is confusing and ambiguous. Reword for clarity; for example: "Compared to non-infested stems, gall-infested stems had lower chlorophyll and carotenoid contents, while enzyme activities were significantly higher."
Revise the abstract to improve the clarity and precision of language. Ensure physiological and biochemical distinctions are articulated clearly and avoid vague generalizations.

Introduction:
Lines 59–61: The logic behind the “chemical arms race” could be better substantiated with specific references. While the authors refer to Roy et al. (2022), additional literature should be incorporated to show the evolutionary coadaptation of gall-inducing insects and their hosts.
Lines 88–93: The botanical classification of Deverra is correct, but overly detailed for this study. Condense this section to maintain focus on the relevance of D. tortuosa as a medicinal plant and its interaction with S. buboniae.
Refocus the introduction to present a stronger rationale for this specific plant-insect system, highlighting the novelty and hypothesis more clearly.

Materials and Methods
Line 127: The sampling description lacks replication and environmental context (e.g., soil type, climate data, plant age).
Lines 133–359: While detailed, many methods rely heavily on classical protocols with “minor modifications” without specifying what those modifications were. For example, in Line 236 and Line 248, the modifications should be explicitly stated.
Line 360: Statistical analysis lacks detail. The use of only t-tests and one-way ANOVA is inadequate for such a broad dataset. Were assumptions of normality and homoscedasticity tested?
Authors must improve reproducibility by stating specific modifications, sample sizes (n), and experimental replication clearly. The statistical section should include multivariate or post-hoc analyses and checks for parametric test assumptions.

Results :
Lines 367–370: The claim of 88% reduction in carotenoids is high; clarify whether this is relative or absolute, and support with statistical significance values.
Lines 386–392: Although nutritional values were compared, the ecological or physiological relevance of such differences in lipid and carbohydrate contents is not discussed.
Line 407–409: Anti-nutritional compound accumulation is interesting, but no reference to thresholds for toxicity or ecological implications is provided.
Authors should contextualize findings more critically—excessive reporting of percentages without biological interpretation weakens the results. Figures should include error bars, sample sizes, and statistical letters or indicators.

Discussion:
Line 451: “First to quantitatively analyze…” – This claim is too strong. There are existing quantitative studies on gall-inducing insects and plant secondary metabolites.
Lines 466–472: The explanation of photosynthetic pigment reduction is supported by citations but lacks data integration. For example, correlating pigment loss with hormone shifts would strengthen the argument.
Lines 473–477: The role of antioxidant enzymes is stated but needs mechanistic depth (e.g., oxidative stress signaling cascades or plant-insect feedback).
Lines 520–538: The discussion on flavonoids and tannins is overly descriptive and cites parallel findings without engaging with possible causes or implications (e.g., are these accumulations plant defense or insect-induced manipulations?).
Authors should critically discuss the mechanistic basis of metabolite shifts and consider multiple biological interpretations (e.g., plant defense vs. insect control). Avoid overstatement of novelty and ensure citation support is balanced.

Conclusion:
Line 578: The phrase “minimal scientific research” is too broad—be precise. There is work on similar systems.
Lines 582–590: The conclusion repeats results without adding synthesis or insight into future research directions. The pharmaceutical claims are speculative and unsupported by bioactivity assays (e.g., no antimicrobial, cytotoxic, or enzyme inhibition tests were performed beyond antioxidant capacity)
Rewrite the conclusion to reflect limitations, suggest functional bioassays for follow-up, and avoid overstating medical potential unless supported by direct evidence.
Also, the manuscript contains frequent grammatical issues (e.g., “increased concentrations of physiology,” “was higher than those in”) and inconsistent phrasing (e.g., “with gall,” “with insect gall,” “non-galled”). I strongly recommend comprehensive English editing by a fluent speaker or a professional service.

Figures and Tables: Figures lack proper annotation and statistical indicators (e.g., p-values, letters for post-hoc tests).
The quality and readability of Figures 3–9 should be improved, especially in color contrast and labeling.
Table 1 and Table 2 should indicate sample sizes and define abbreviations used (e.g., Q eq/g, RT eq/g).

My Recommendation: Major Revision

Given the scientific potential of the dataset and the novel pairing of species, this manuscript shows promise. However, significant revisions are necessary in the experimental detail, language quality, statistical analysis, figure clarity, and most importantly, the critical discussion of the findings. I will reconsider my recommendation after the authors provide a thoroughly revised version addressing all points listed above.

Reviewer 2 ·

Basic reporting

The authors have investigated the biochemical interactions between Deverra tortuosa and the gall-inducing insect Schizomyia buboniae, focusing on a wide range of primary and secondary metabolites, antioxidant activities, and hormonal changes. This is a scientifically valuable and interesting study that contributes to our understanding of plant-insect relationships. However, several major and minor issues need to be addressed to improve the clarity, structure, and scientific rigor of the manuscript before it can be considered for publication.

Experimental design

Abstract
1. The abstract does not report specific numerical data, statistical results (e.g., % increases, p-values), or magnitude of changes. Abstracts should reflect the strength and significance of the results numerically wherever possible.

2. The aim sentence is long, repetitive, and uses imprecise language.

3. “This study aimed to investigate the interaction between gall inducer by Schizomyia buboniae and Deverra tortuosa, including the physiology, phytochemical, antioxidants, and antinutrient content…“Clarify the objective in a concise sentence. Suggested rewrite: This study investigated the impact of Schizomyia buboniae-induced galls on physiological traits, phytochemical profiles, antioxidant capacity, and antinutrient levels in Deverra tortuosa stems.

4. Repeats "D. tortuosa stems with galls" and "compared to stems without galls" excessively. “Condense repeated phrases and group findings logically. For example: ‘Stems with galls showed significantly higher levels of antioxidants, phytochemicals, and antinutrients compared to non-galled stems.’”

5. Unclear or Misleading Comparisons, “...as being compared with that of the non-infested stems with galls...” “Ensure consistent terminology: refer to groups clearly as ‘galled’ vs. ‘non-galled’ stems. Avoid confusing phrasing such as ‘non-infested stems with galls’.”
6. Final statement is vague and overly ambitious: “...has the potential to serve as a natural resource for innovative pharmaceutical and medical applications.” “Be specific and avoid overgeneralization unless supported by results. For instance: ‘These findings suggest that gall-induced tissues of D. tortuosa may be valuable sources of bioactive compounds for pharmaceutical applications.’”

Introduction
1. The opening sentence is not clear: “Galls are nodules or protuberances that develop when another organism stimulates plant tissue, leading to abnormal differentiation and accelerated cell division.” Split into two clearer sentences. Ex. “Plant galls are abnormal outgrowths that form when certain organisms stimulate plant tissues. This stimulation causes accelerated cell division and tissue differentiation.”

2. Insects that cause galls are highly skilled at altering the phenotypes of their hosts. What is the meaning of “altering phenotypes? It means “Gall-inducing insects can manipulate plant growth and chemical profiles to favor their development. They alter host plant phenotypes by modulating phytochemical pathways, inducing tissue growth, and enhancing defense responses.”?

3. “Within the gall midge subfamily, animalia is the newest…” "Animalia" is a kingdom, not a subfamily. Please check it again.

4. The aim of this study should be clearly stated, including the identification of the research gap and the specific objectives. Additionally, the grammar throughout the manuscript should be improved.

Validity of the findings

Methods
How to prepare the sample before measurements of enzyme activities, phenolics, tannins, flavonoids, flavonols?
Results

1. Photosynthetic Pigments
The sentence “...declined by about 63, 14, 44 and 88 % less...” Reword the awkward “less than that of” structure. Were these differences statistically significant? Add: “(P < 0.05)” where applicable.

2. Enzyme Activities
Clarify which enzyme each % increase refers to?

3. The results should include P-values consistently where comparisons are made. Ensure sample size (n) is stated somewhere (“means ± SD of 3 replicates”, etc.). Ensure Figures and Tables are clearly referenced in logical order. Add statistical tests used for each dataset if not already mentioned in the methods.

Discussion
1. “This study is the first to quantitatively analyze... secondary metabolites of the plants.” Instead of saying “first,” specify what exactly is novel: scope, method, or combination of parameters.

2. Gall-inducing species can change the host plants' physiology... decline in the photosynthesis rate [69]. Please streamline and combine points of group-related mechanisms (e.g., pigment biosynthesis interruption and sink theory).

3. In plants that are fed by insects,... deadly to insects [74]. Please clarify the function to differentiate between defense signaling vs. direct toxicity.

4. Plant cells contain low or high concentrations... stems with galls. Discuss each hormone briefly, such as IAA, GA3, and ABA. And clarify insect role, for example, “Increases in IAA, GA3, and ABA suggest a manipulation of host hormone pathways, potentially orchestrated by gall-inducing larvae [79–81]. This hormonal reprogramming likely facilitates gall initiation and development.”

5. You should separate subsections or paragraph breaks for each nutrient class for clarity. My suggestions: Carbohydrates: Emphasize energy redistribution. Proteins: Link to metabolic activity or stress response. Lipids: Mention their role in structural changes or membrane dynamics.

6. The authors currently lack a closing paragraph. Please add the following conclusion, relating between the presence of Schizomyia buboniae-induced galls and alters the biochemical landscape of D. tortuosa stems potential use applications in galled plant tissues.

Additional comments

Others:
- Check Spelling
- Check Grammar
- Add statistical tests
- Add means ± SD, replication
- Figure 3-5 are not clear.
- Figure 6-9, please add the error bar.

·

Basic reporting

-

Experimental design

-

Validity of the findings

-

Additional comments

This manuscript investigation is thoughtfully constructed. In addition, the methodology and results are both well-emphasized and transparent. I advise the authors to respond to the following criticisms appropriately.

1. Ensure the abstract is more precise and clearly highlights the key outcomes of the study.

2. Carefully review and correct all genera and species names throughout the manuscript to ensure they follow proper scientific nomenclature and formatting conventions.

3. Improve the overall English language quality of the manuscript.

4. Identify and correct all minor typographical, grammatical, and syntactical errors.

5. Strengthen the introduction and discussion sections, particularly in relation to the characterization and in vitro inhibition assay of antioxidant studies.

6. Carefully revise the In Vitro Antioxidant Activity section.

7. Improve the quality of all images and figures in the manuscript.

8. Further emphasize the novelty of the research.

9. Revise the conclusion to better reflect the study’s key findings, implications, and potential future directions.

---

## Round 0.2 · accepted · Accept

· Academic Editor

Accept

We have received evaluations from two expert reviewers who commented that you have incorporated all of their suggestions and modified the manuscript accordingly. I am pleased to say that I am recommending that your manuscript be accepted for publication in PeerJ.

Reviewer 1 ·

Basic reporting

.

Experimental design

.

Validity of the findings

.

Additional comments

.

Reviewer 2 ·

Basic reporting

Good.

Experimental design

Good.

Validity of the findings

Good

Additional comments

The authors have incorporated the reviewers’ comments, clarifying the report, results, validation, and discussion.